# Containment of a healthcare-associated COVID-19 outbreak in a university hospital in Seoul, Korea: A single-center experience

**Sei Won Kim**[1], **Sung Jin Jo**[2], **Heayon Lee**[1], **Jung Hwan Oh**[3], **Jihyang Lim**[2], **Sang Haak Lee**[1], **Jung Hyun Choi**[4], **Jehoon Lee**[2]*

**1** Division of Pulmonary, Critical Care and Sleep Medicine, Department of Internal Medicine, Eunpyeong St. Mary's Hospital, College of Medicine, The Catholic University of Korea, Seoul, Republic of Korea, **2** Department of Laboratory Medicine, Eunpyeong St. Mary's Hospital, College of Medicine, The Catholic University of Korea, Seoul, Republic of Korea, **3** Division of Gastroenterology, Department of Internal Medicine, Eunpyeong St. Mary's Hospital, College of Medicine, The Catholic University of Korea, Seoul, Republic of Korea, **4** Division of Infectious Diseases, Department of Internal Medicine, Eunpyeong St. Mary's Hospital, College of Medicine, The Catholic University of Korea, Seoul, Republic of Korea

☯ These authors contributed equally to this work.
* lyejh@catholic.ac.kr

**Data Availability Statement:** All relevant data are within the manuscript.

## Abstract

### Background

Our hospital experienced the first healthcare-associated COVID-19 outbreak in Seoul at the time the first COVID-19 cases were confirmed in Korea. The first confirmed COVID-19 patient was a hospital personnel who was in charge of transferring patients inside our hospital. To contain the virus spread, we shutdown our hospital, and tested all inpatients, medical staff members, and employees.

### Methods

We retrospectively analyzed the results of SARS-CoV-2 RT-PCR testing according to the contact history, occupation, and presence of respiratory symptoms. Closed-circuit television (CCTV) was reviewed in the presence of an epidemiologist to identify individuals who came into contact with confirmed COVID-19 patients.

### Results

A total of 3,091 respiratory samples from 2,924 individuals were obtained. Among 2,924 individuals, two inpatients, and one caregiver tested positive (positivity rate, 0.1%). Although all confirmed cases were linked to a general ward designated for pulmonology patients, no medical staff members, medical support personnel, or employees working at the same ward were infected. Contact with confirmed COVID-19 cases was frequent among inpatients and medical support personnel. The most common contact area was the general ward for pulmonology patients and medical support areas, including clinical and imaging examination rooms. Finally, the total number of hospital-associated infections was 14, consisting of four diagnosed at our hospital and ten diagnosed outside the hospital.

**Funding:** The author(s) received no specific funding for this work.

**Competing interests:** The authors have declared that no competing interests exist.

## Conclusions

The robust control of the COVID-19 outbreak further minimized the transmission of SARS-CoV-2 in the hospital and local communities. However, there was also a debate over the appropriate period of hospital shutdown and testing of all hospital staff and patients. Future studies are required to refine and establish the in-hospital quarantine and de-isolation guidelines based on the epidemiological and clinical settings.

## Introduction

In December 2019, a novel coronavirus disease (COVID-19) was first reported in Wuhan, Hubei Province, China [1, 2]. Since then, human-to-human transmission of the novel coronavirus has been confirmed [3] and has caused serious illness and death [4]. Following the COVID-19 outbreak in China, severe acute respiratory syndrome coronavirus 2 (SARS-CoV-2), the virus responsible for COVID-19, has spread in many countries across the world, including Korea, Iran, European countries, and the USA [5].

In Korea, the number of COVID-19 patients who were confirmed with real-time reverse-transcription polymerase chain reaction (RT-PCR) assay was 10,564, while 222 deaths were reported until April 14, 2020 [6]. Among the COVID-19 patients, 14 patients were linked to a single university hospital. These were the first COVID-19 cases resulting from healthcare-associated infections (HAI) in Seoul. Hence, the Seoul government announced the closure of the hospital. Moreover, outpatient admission was temporarily stopped for approximately two weeks. Inpatients who had come into contact with confirmed COVID-19 patients were isolated inside the hospital. A total of 2,924 people, including hospitalized patients, medical/para-medical staff, employees, and caregivers, were all tested for SARS-CoV-2 using RT-PCR assay.

In this study, we report our experience from our COVID-19 cases, as well as discuss the necessity of widespread SARS-CoV-2 RT-PCR testing and hospital shut down, particularly in major hospitals, to prevent HAI.

## Materials and methods

### Hospital overview and study setting

Our hospital (Eunpyeong St. Mary's hospital, The Catholic University of Korea) is an 808-bed university hospital located in Eunpyeong District, northwest Seoul, Korea. The hospital opened in April 2019. The main building of the hospital has 17 floors above ground and seven floors below ground. The hospital receives approximately 2000 to 3000 outpatient visits and 150 to 200 patient visits to the emergency room daily. The hospital has nine floors for inpatients, which contains two general wards. Each general ward has nine rooms with four beds each, and two rooms with a single bed. The 4-bed rooms have 2 m space between beds, which are separated by curtains for privacy (Fig 1).

In this study, we retrospectively analyzed the results of SARS-CoV-2 RT-PCR testing, contact history, and presence of respiratory symptoms in a single center with a healthcare-associated COVID-19 outbreak. We also reviewed data from epidemiological surveys, from the Korea Centers for Disease Control and Prevention (KCDC) and the infection control unit of our hospital. The COVID-19 prevention measures before and after the healthcare-associated outbreak were also reviewed. This study was approved by the Institutional Review Board of Eunpyeong St. Mary's hospital (PC20RASI0040).

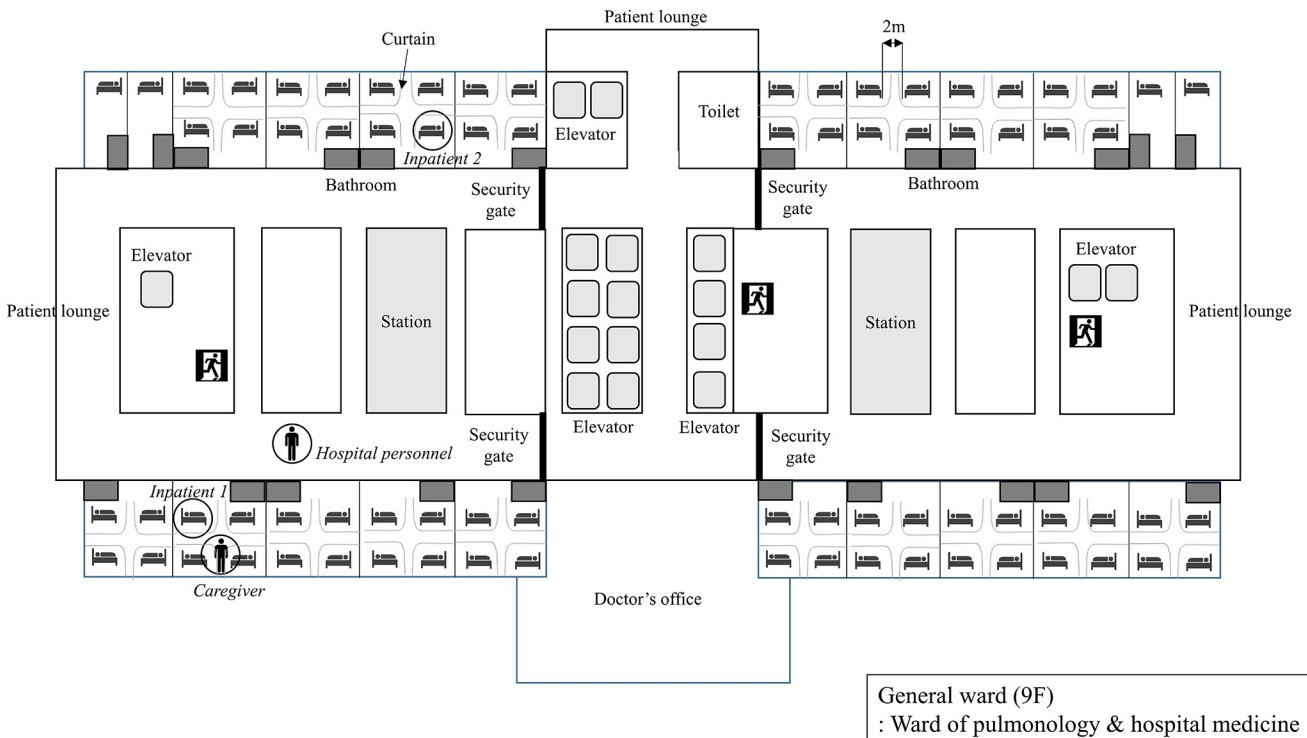

**Fig 1. Schematic diagram of the general ward located on the 9ᵗʰ floor and location of confirmed COVID-19 patients.**

## Hospital COVID-19 prevention measures before the healthcare-associated outbreak

We reviewed the history of patients to assess whether they visited China or other high-risk countries within two weeks prior to the outbreak of healthcare-associated COVID-19, or if they came into contact with confirmed COVID-19 cases. After the initial assessment, patients without fever or respiratory symptoms had no specific restrictions inside the hospital. Patients with fever (>37.5 ˚C) or respiratory symptoms were transferred to the triage room, which is located in a different part of the hospital to that of the infectious diseases department and the emergency room. After obtaining samples for RT-PCR testing, patients were sent home for self-isolation until the results were made available. If the results were positive, patients were referred to government-designated hospitals. Medical staff in the triage room used level-D personal protective equipment (PPE) and everyone in the hospital was encouraged to wear masks and follow hand hygiene practices.

## The first confirmed COVID-19 case in the hospital

The first confirmed COVID-19 case was a 35-year-old man who was in charge of transferring patients inside our hospital between February 2 and 17. On February 17, 2020, he visited the clinic due to a one-week history of fever, cough, and myalgia. Although he had no history of travel or close contact with confirmed COVID-19 cases, chest X-ray showed ground-glass opacities in both lower lobes. Considering the possibility of COVID-19, RT-PCR confirmation testing was performed on February 20, 2020. After SARS-CoV-2 infection was confirmed, the Seoul city government announced the closure of the hospital on February 21, 2020, to prevent

a healthcare-associated outbreak. Outpatient clinics and the emergency room were also closed, and new patient admissions were stopped. Individuals inside the hospital who had contact history, fever, or respiratory symptoms, were closely monitored.

### Samples collection and test systems

In total, 3,091 respiratory samples from 2,924 people were obtained from February 21 to February 28. Two thousand one hundred seventy-one samples were processed and tested in the department of Laboratory Medicine of our hospital, while 920 samples were analyzed by a commercial laboratory (SamKwang Medical Laboratories, Korea).

Sputum and combined nasopharyngeal (NP) and oropharyngeal (OP) swabs were collected from patients with acute respiratory symptoms and purulent manifestations. Combined NP and OP swabs were also collected from asymptomatic individuals. NP and OP swabs were collected in the T-SWAB TRANSPORT™ UTM (Noble Biosciences, Korea) while sputum specimens were collected in 50cc Falcon tubes. Sputum samples were prepared in phosphate buffer solution. QIAamp DSP viral RNA mini kit (Qiagen GmbH, Hilden, Germany) with QIAcube system (Qiagen), as well as NX-48 viral NA kit (Genolution, Korea) with Nextractor NX-48 system (Genolution) were used for RNA extraction. Nucleic acid was extracted according to the manufacturer's instructions. SARS-CoV-2 nucleic acid was amplified by real-time RT-PCR using the PowerCheckTM 2019-nCoV Real-time PCR Kit (Kogenebiotech, Korea). ABI 7500 (Applied Biosystems, USA) real-time PCR system was used for the amplification of $E$ and $RdRp$ genes of the SARS-CoV-2 virus. In SamKwang Medical Laboratories, the NX-48 viral NA kit (Genolution) with Nextractor NX-48 system (Genolution) was used for RNA extraction, while PowerCheckTM 2019-nCoV Real-time PCR Kit (Kogenebiotech) and CFX96 (Bio-Rad, USA) were used for the amplification of viral genes. The RT-PCR included 40 cycles of amplification. SARS-CoV-2 infection was defined as the detection of both target RNAs under 35.0 cycles of threshold (CT).

### Contact history

The results of SARS-CoV-2 RT-PCR were analyzed according to occupation, presence of the respiratory symptoms, and contact history. The admission department, floor, and room location were also assessed. Contact was defined as presence in the same room with COVID-19 confirmed patients, or in the same outpatient clinic or examination room, 30 minutes before and after COVID-19 confirmed patients. Moreover, closed-circuit television (CCTV) was reviewed in the presence of an epidemiologist, and people who had been within 2 m of confirmed patients were considered individuals with contact history.

## Results

### Patients with healthcare-associated COVID-19 infection

After the hospital staff member responsible for transporting patients was confirmed as the first COVID-19 case, people with contact history, fever, or respiratory symptoms were tested for SARS-CoV-2 infection with RT-PCR (Fig 2). Among the admitted patients, patient with pneumonia who was receiving treatment was confirmed as the 2[nd] COVID-19 patient (Fig 3). Chest CT showed that the patient had multiple, ground-glass opacities in both lungs. This patient was not previously tested for COVID-19, as the patient had no history travel to China or other high-risk countries, and had no close contact with confirmed patients. Due to a contact history with this inpatient, the first diagnosed medical staff was considered as nosocomial COVID-19 infection. Additionally, one caregiver who was in the same room with the 2[nd] COVID-19

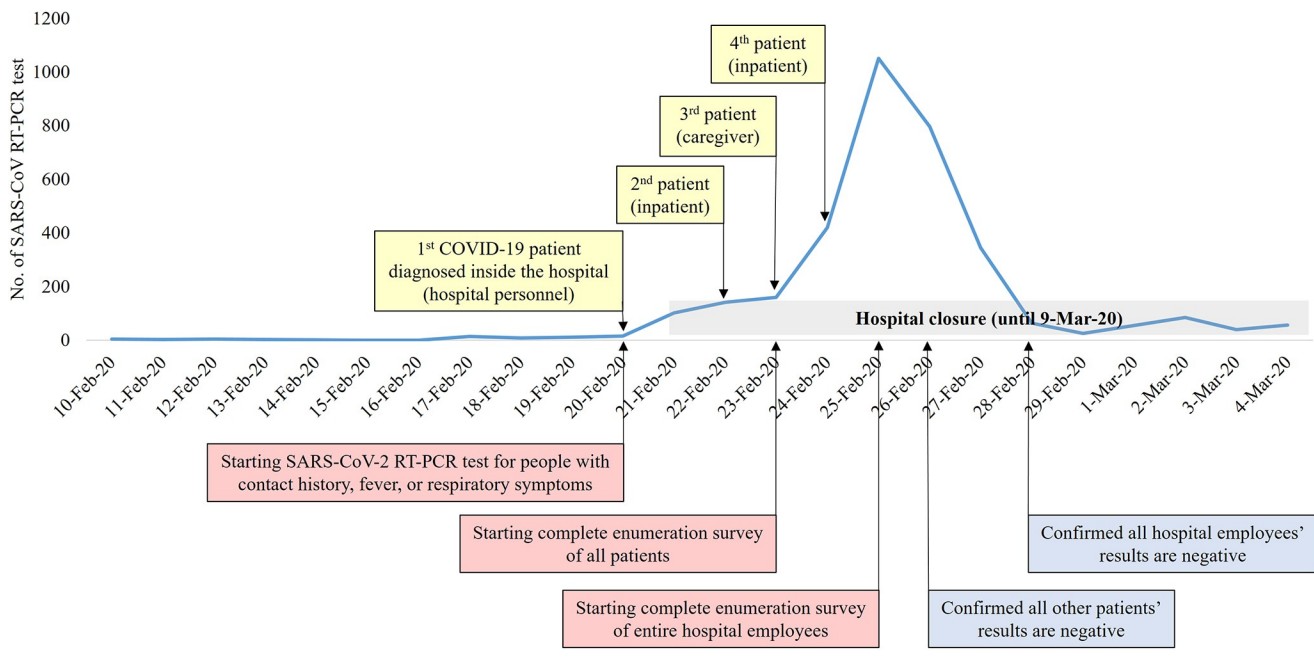

**Fig 2. Timeline of healthcare-associated COVID-19 outbreak and number of SARS-CoV-2 RT-PCR tests performed.**

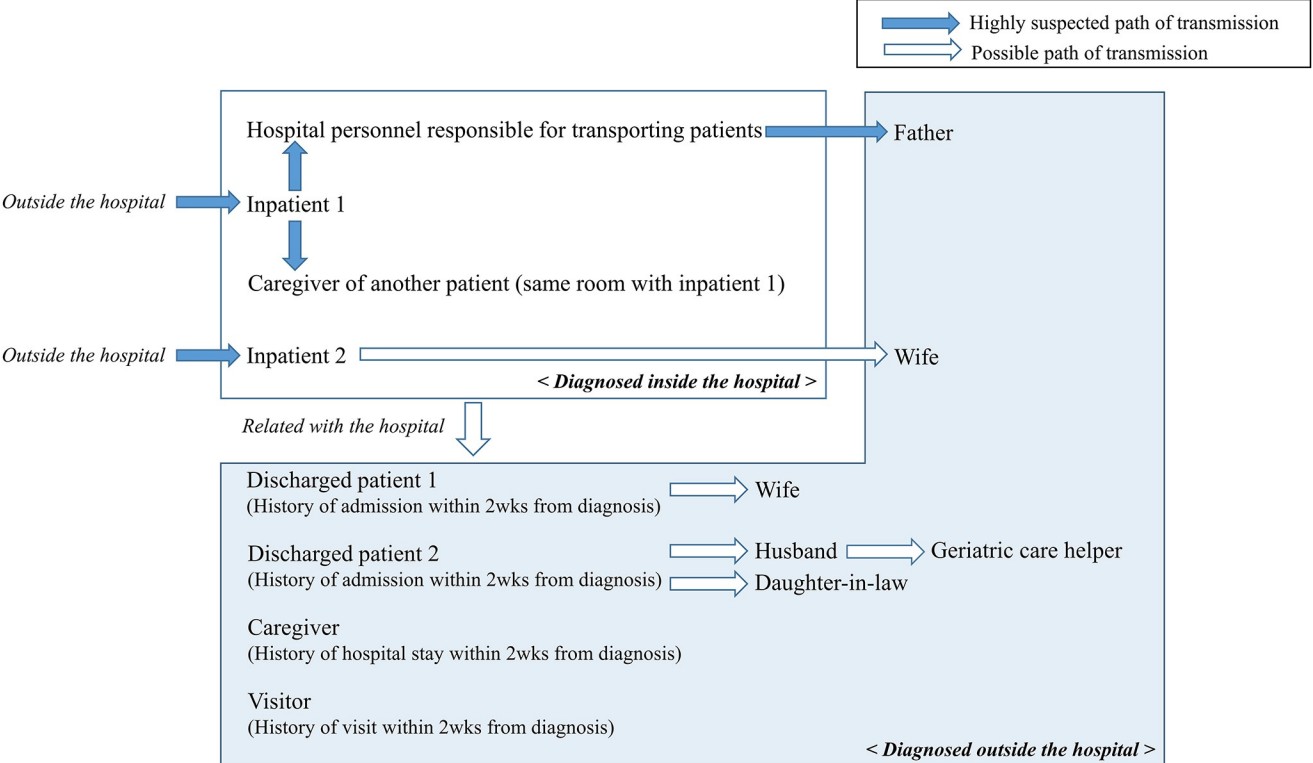

**Fig 3. Estimated transmission routes of 14 cases of healthcare-associated SARS-CoV-2 infection.**

patient also tested positive for SARS-CoV-2. The caregiver confirmed as COVID-19 was also considered as nosocomial COVID-19 infection. As the nosocomial COVID-19 infection continued to be discovered, the hospital started complete enumeration survey of all patients (Fig 2). The fourth COVID-19 patient was also an inpatient with pneumonia such as the second COVID-19 patient. On February 25, the hospital decided to perform a complete enumeration survey of all medical staff and employees to help control the healthcare-associated COVID-19 outbreak.

## Quarantine after the healthcare-associated COVID-19 outbreak and complete enumeration survey of all inpatients, medical staff, and employees

As this was the first major healthcare-associated transmission cluster reported in Seoul, the city government imposed strict measures to control the spread of the virus. The entire outpatient clinic and emergency rooms were temporarily closed for two weeks, as per the guidelines set during the 2015 Middle East Respiratory Syndrome (MERS) outbreak [7]. The entire hospital was thoroughly cleaned and disinfected. Inpatients who had had no contact with confirmed COVID-19 patients and had no symptoms were discharged, while inpatients who had had contact with confirmed COVID-19 patients were quarantined in single rooms for two weeks. A complete enumeration survey was conducted from January 23 to January 28 to prevent further spread of the healthcare-associated infection by eliminating the possibility of asymptomatic transmission [8]. Two thousand nine hundred twenty-four inpatients and employees (213 doctors, 901 nurses, 271 medical support staff, 952 hospital employees, 494 inpatients, 87 guardians and caregivers, and 11 volunteers) had undergone SARS-CoV-2 testing by real-time RT-PCR (Table 1). The employees who had contact with confirmed COVID-19 patients self-isolated for two weeks. Two patients and one caregiver tested positive for SARS-CoV-2. The overall COVID-19 positivity rate in the complete enumeration survey ($100 \times$ positive tests/ total number of tests conducted) was 0.1%. Fifty-one individuals were re-tested more than twice to monitor the progression of respiratory symptoms.

**Table 1. Clinical characteristics of individuals who underwent SARS-CoV-2 real-time RT-PCR testing during Feb. 21 –Feb. 28, 2020, in Eunpyeong St. Mary's Hospital after the 1st COVID-19 patient was confirmed.**

| Groups | No. | Age | Sex (male/ female) | No. of COVID-19 patients (RT-PCR confirmed) | No. of people resampled and re-tested | No. of people with respiratory symptoms |
|---|---|---|---|---|---|---|
| Doctors | 213 | 40.9 ± 8.8 | 129 / 84 | 0 | 26 | 2 |
| Nurses | 901 | 31.0 ± 7.8 | 42 / 859 | 0 | 4 | 7 |
| Medical support personnel | 271 | 34.1 ± 8.5 | 130 / 141 | 0 | 0 | 3 |
| Employees | 952 | 44.4 ± 12.1 | 368 / 584 | 0 | 1 | 5 |
| Inpatients | 489 | 61.8 ± 19.2 | 229 / 260 | 2 | 20 | 104 |
| Guardians and caregivers | 87 | 59.4 ± 11.8 | 20 / 67 | 1 | 0 | 2 |
| Volunteers | 11 | 47.7 ± 9.8 | 2 / 9 | 0 | 0 | 0 |
| **Total** | **2,924** | **42.5 ± 16.3** | **920 / 2,004** | **3** | **51** | **123** |

SARS-CoV-2 real-time RT-PCR: Severe acute respiratory syndrome-coronavirus-2 real-time reverse transcriptase polymerase chain reaction, COVID-19: Coronavirus disease 2019

**Table 2. Occupation and number of individuals who had contact with four confirmed COVID-19 patients.**

| Groups | 1st patient (Hospital staff) | 2nd patient (Inpatient, 9GW) | 3rd patient (Caregiver, 9GW) | 4th patient (Inpatient, 9GW) | 1st– 4th patient |
|---|---|---|---|---|---|
| Doctors | 1 | 10 | 8 | 4 | 20* (9.4%)[†] |
| Nurses | 0 | 31 | 36 | 37 | 72* (8.0%)[†] |
| Medical support personnel | 2 | 22 | 8 | 14 | 35* (12.9%)[†] |
| Employee | 2 | 10 | 11 | 7 | 19* (2.0%)[†] |
| Inpatients | 47 | 34 | 41 | 17 | 138* (28.2%)[†] |
| Caregivers | 0 | 4 | 2 | 0 | 6 (6.8%)[†] |
| Volunteer | 0 | 0 | 0 | 0 | 0 (0%)[†] |
| Total | 52 | 111 | 106 | 79 | 290* (9.9%) |

COVID-19: Coronavirus disease 2019, SARS-CoV-2 real-time RT-PCR: Severe acute respiratory syndrome-coronavirus-2 real-time reverse transcriptase polymerase chain reaction, GW: general ward

*Multiply contacted person was counted as one.

[†] Percentage = 100 × number of people who had contact with confirmed patients/ total number of people in each group

## Contact history with confirmed COVID-19 patients

After the first case was reported, epidemiologists from KCDC and the infection control unit of our hospital reviewed electronic medical charts, CCTV, and personal movements to identify individuals with potential contact with confirmed COVID-19 patients. An additional three confirmed cases were identified, and the contact list was updated. The number of people who underwent SARS-CoV-2 testing with real-time RT-PCR and had contact with COVID-19 confirmed patients is shown in Table 2. The overall proportion of people who came into contact with confirmed COVID-19 patients was 9.9%. Most of these individuals were inpatients (28.2%) who had stayed on the same ward with confirmed COVID-19 patients. Contact with confirmed COVID-19 cases was also frequent among medical support personnel (12.9%), including staff from the radiologic department, rehabilitation unit, and phlebotomists. Three confirmed COVID-19 patients (2nd, 3rd, 4th patients) stayed on 9GW. Four caregivers, 40 healthcare workers, and 12 inpatients came into contact with confirmed COVID-19 patients on 9GW. The most common area of contact with confirmed COVID-19 patients was the 9GW and medical support area, such as the clinical or imaging examination room (Table 3).

## Surveillance of people with contact history with confirmed COVID-19 patients

Discharged patients, caregivers, and healthcare workers who had left before the hospital closed were grouped according to the possibility of COVID-19 exposure, based on electronic medical charts, CCTV, and personal movements. Four hundred and seventy-nine people were determined as high-risk, including 58 caregivers, 11 healthcare workers, and 410 discharged patients, all of whom were notified by phone. They were also examined for respiratory symptoms and advised to stay home and avoid contact with other people even if they had no respiratory symptoms. Two discharged patients, one caregiver and one visitor were later diagnosed with COVID-19 at a different hospital or community health center (Fig 3). As additional patients were confirmed, the number of exposed people increased to 1,215; these individuals received advice as per the KCDC COVID-19 guidelines. Finally, the total number of hospital-associated infections was 14, consisting of four diagnosed at our hospital and ten diagnosed outside the hospital.

**Table 3. Place of contact and number of individuals who came into contact with four confirmed COVID-19 patients.**

| Places | 1st patient (Hospital staff) | 2nd patient (Inpatient, 9GW) | 3rd patient (Caregiver, 9GW) | 4th patient (Inpatient, 9GW) | 1st– 4th patient |
|---|---|---|---|---|---|
| Out-patient clinic | 3 | 4 | 3 | 5 | 13* |
| Emergency room | 0 | 10 | 0 | 0 | 10 |
| Medical support area | 7 | 63 | 41 | 33 | 131* |
| Intensive care unit | 2 | 0 | 0 | 0 | 2 |
| 7GW | 5 | 0 | 0 | 0 | 5 |
| 8GW | 0 | 2 | 5 | 18 | 25 |
| 9GW | 8 | 33 | 37 | 22 | 60* |
| 10GW | 2 | 1 | 21 | 1 | 23* |
| Others | 25 | 0 | 1 | 0 | 26 |
| Total | 52 | 111* | 106* | 79 | 290* |

COVID-19: Coronavirus disease 2019, SARS-CoV-2 real-time RT-PCR: Severe acute respiratory syndrome-coronavirus-2 real-time reverse transcriptase polymerase chain reaction, GW: general ward

*Multiply contacted person was counted as one.

## Discussion

Since the report of the first COVID-19 cases in Wuhan, China, 1,812,734 confirmed cases and 113,675 deaths have been reported worldwide until April 14, 2020 [9]. Severe symptoms develop in approximately 14% of COVID-19 patients, and the overall mortality is around 2% of confirmed COVID-19 cases [10]. Advanced age, development of severe symptoms, and comorbidities are the primary risk factors associated with COVID-19 mortality [11, 12].

As there are no effective treatments for COVID-19, prevention of further virus spread is the only way to control the COVID-19 pandemic [13]. Prevention measures are particularly important in hospitals, where numerous elderly patients and individuals with comorbidities are found. In 2015, nosocomial MERS outbreaks were reported in Korea, caused by contacts with infected outpatients and inpatients [14]. Although strict precaution and prevention measures were followed according to the KCDC guidelines, 14 healthcare-associated COVID-19 cases were reported in our hospital, including 4 people diagnosed in-hospital and 10 people diagnosed outside of the hospital. Compared to small hospitals or care units, the impact of healthcare-associated infections can be detrimental in large university hospitals with numerous visitors and severely ill patients.

The first confirmed COVID-19 patient in our hospital was a hospital personnel responsible for transporting patients. However, the only person who came into contact with him and was later confirmed with COVID-19 was his father. The second confirmed COVID-19 patient (inpatient with pneumonia) showed high viral RNA load (CT values in RT-PCR: *E* gene, 17.19; *RdRp* gene, 17.64). Despite the high viral load, only two direct transmissions could be identified. Asymptomatic transmission of COVID-19 has also been reported [15, 16], and viral loads can be high as symptomatic carriers [17]. Therefore, to control the spread of the virus, we tested all people in the hospital to identify asymptomatic and undiagnosed individuals. Despite extensive exposure to four confirmed patients, only a few HAI were detected.

Several reasons could explain the minimal spread of SARS-CoV-2 in our hospital. Importantly, Korea experienced an outbreak of MERS in 2015; hence, nearly all hospitals have implemented infection prevention guidelines [7]. Immediately after the outbreak of COVID-19 in China, the Korean government imposed prevention measures [18], and the infection control unit of our hospital restricted hospital visitors with respiratory symptoms or travel history. As

per the KCDC recommendations [19], masks were recommended for all patients and medical staff, and the importance of hand-washing was emphasized. Furthermore, our hospital was newly built with good ventilation facilities. Distance between the beds in multi-bed rooms was over 2 m. We know from the SARS outbreak that distance between beds ≤ 1m was a significant risk factor associated with healthcare-associated SARS transmission [20]. Curtains were installed between the beds, most of which were kept closed for privacy. This could have greatly contributed to the containment of the virus spread [21]. Additionally, the maximum number of beds in a room was four, which is lower than other university hospitals in Seoul. Hospital closure, inpatient isolation in single rooms, self-isolation of individuals with contact history for two weeks, and complete enumeration survey of all inpatients, medical staff, and employees regardless of contact history or development of respiratory symptoms, further minimized the transmission of SARS-CoV-2 in the hospital and local communities. The robust control of the nosocomial COVID-19 outbreak reassured the remaining inpatients and local community that the risk of in-hospital transmission was low. However, the measures taken were time-consuming, laborious, and expensive.

Complete enumeration could be achieved due to the rapid installation of new molecular equipment. Before the nosocomial outbreak, one ABI 7500 (Applied Biosystems) real-time PCR system and two QIAcube (Qiagen) preparation systems were available in the Department of Laboratory Medicine to test for SARS-CoV-2 and other pathogens. To increase SARS-CoV-2 RT-PCR testing capacities during the nosocomial outbreak, we installed new viral RNA preparation systems and one ABI 7500 system on February 24th. Comparison between Nextractor NX-48 systems and QIAcube systems, as well as between two ABI systems, were performed. The test capacity of SARS-CoV-2 RT-PCR testing increased to approximately 800 tests per day after the installation of the new equipment. Including the commercial laboratory tests, we could test approximately 1,000 samples in 24 hours. In addition to the introduction of new equipment, the working hours and shifts of laboratory medical technicians were adjusted to allow for 24-hour testing.

The infection control unit and every department of our hospital continuously monitored inpatients and employees for fever and respiratory symptoms. Fifty-one individuals underwent SARS-CoV-2 RT-PCR testing more than twice to monitor the progress of respiratory symptoms. Moreover, for some individuals, several respiratory specimens were tested to account for the incubation period of SARS-CoV-2, which is estimated to be 5.1 days [22]. Moreover, SARS-CoV-2 RT-PCR testing was repeated prior to patient de-isolation; however, considering the cost, this may not be possible in all hospitals [23].

Our hospital was reopened on March 9th, 17 days after the closure and, until now, no additional COVID-19 cases have been confirmed. The duration of the hospital closure was based on the Seoul government guidelines established during the MERS outbreak [7]. According to "Management of medical center with COVID-19 confirmed patients" guidelines established by the Central Disaster and Safety Countermeasures Headquarters of Korea, hospitals should close when the risk of transmission is high [24]. On March 9th, the Korean Medical Association announced that 17 days of closure would be a laborious administrative process and suggested that the old guidelines based on the MERS outbreak should be revised. During the hospital closure, the existing patients could not receive treatments, and some were denied care elsewhere. Due to the complete hospital shutdown, patients with scheduled chemotherapy, radiation therapy, hemodialysis, emergency operation, or scheduled births faced health challenges.

In conclusion, the robust control of the COVID-19 outbreak further minimized the transmission of SARS-CoV-2 in the hospital and local communities. However, there was also a debate over the appropriate period of hospital shutdown and testing of all hospital staff and

patients. Future studies are required to refine and establish the in-hospital quarantine and de-isolation guidelines based on the epidemiological and clinical settings.

## Acknowledgments

We greatly appreciate the support from Soon-Yong Kwon, Seung-Hye Choi, Rev. Fabian Park Chang Yeob, Jae-Taek Hong, Seung Eun Jung, infection control unit and emergency headquaters of Eunpyeong St. Mary's Hospital. We also thank all the members of the Eunpyeong St. Mary's Hospital for their efforts and devotion during the crisis of COVID-19.

## Author Contributions

**Conceptualization:** Sei Won Kim, Sung Jin Jo, Jung Hwan Oh, Jihyang Lim, Jehoon Lee.

**Data curation:** Sung Jin Jo, Jihyang Lim, Jehoon Lee.

**Formal analysis:** Sei Won Kim, Sung Jin Jo.

**Investigation:** Sei Won Kim, Sung Jin Jo, Heayon Lee.

**Methodology:** Sei Won Kim, Sung Jin Jo.

**Supervision:** Jehoon Lee.

**Writing – original draft:** Sei Won Kim, Sung Jin Jo.

**Writing – review & editing:** Jung Hwan Oh, Jihyang Lim, Sang Haak Lee, Jung Hyun Choi, Jehoon Lee.

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
