## [Decision Letter · Decision Letter 0]

22 Jun 2020

PONE-D-20-13049

Containment of a healthcare-associated COVID-19 outbreak in a university hospital in Seoul, Korea: a single-center experience

PLOS ONE

Dear Dr. Lee,

Thank you for submitting your manuscript to PLOS ONE. After careful consideration, we feel that it has merit but does not fully meet PLOS ONE’s publication criteria as it currently stands. Therefore, we invite you to submit a revised version of the manuscript that addresses the points raised during the review process.

We look forward to receiving your revised manuscript.

Kind regards,

Xia Jin, MD, PhD

Academic Editor

PLOS ONE

Journal Requirements:

Reviewers' comments:

Reviewer's Responses to Questions

**Comments to the Author**

1. Is the manuscript technically sound, and do the data support the conclusions?

Reviewer #1: Partly

2. Has the statistical analysis been performed appropriately and rigorously? 

Reviewer #1: Yes

3. Have the authors made all data underlying the findings in their manuscript fully available?

Reviewer #1: Yes

4. Is the manuscript presented in an intelligible fashion and written in standard English?

Reviewer #1: Yes

5. Review Comments to the Author

Reviewer #1: This paper illustrates actions taken by south Korea (hereafter Korea) in the case of the first COVID-19 confirmed cases in a university hospital. Korea is one of the countries managing COVID-19 under control without major shutdown. Therefore, their experience is unique and important for medical communities throughout the world.

The hospital identified contact individual with the confirmed patients based on the following criteria: Stayed in the same room within 30 min. interval with confirmed patients including outpatient clinic or examination room. stayed within 2 m of confirmed patients. 2,924 inpatients and employees were tested for COVID-19. Everyone with contact history even without symptom was quarantined for 14 days including medical staff. Two discharged patients and one caregiver were tested positive. The same procedure for each newly confirmed case was taken.

1.1.This paper is not written clearly. The procedures taken is not easy to follow and confusing. For example, page 16 “14 healthcare-associated COVID-19 cases were reported in our hospital.” This number looks like including subsequent confirmed cases related to the hospital illustrated in Figure 2. Without clean explanation this sentence is confusing.

2.This paper conclude hospital shutdown for extended period was not necessary and testing of all hospital staff and inpatients without risk assessment are time-consuming. However, the paper does not provide enough scientific evidence how this conclusion is reached. Low confirmed probability ex-post is not the evidence you can use since the risk of outbreak is still exist.

Just explaining the action taken by Korea is meaningful.

6. PLOS authors have the option to publish the peer review history of their article (what does this mean?). If published, this will include your full peer review and any attached files.

Reviewer #1: Yes: Beomsoo Kim

---

## [Author Response · Author response to Decision Letter 0]

14 Jul 2020

We really appreciate that the reviewer has provided valuable comments on how to improve our research. We fully understood the points from the reviewer and revised the manuscript.

< Reviewer #1 >

1. This paper is not written clearly. The procedures taken is not easy to follow and confusing. For example, page 16 “14 healthcare-associated COVID-19 cases were reported in our hospital.” This number looks like including subsequent confirmed cases related to the hospital illustrated in Figure 2. Without clean explanation this sentence is confusing.

: Thank you very much for the comment. Fourteen healthcare-associated COVID-19 cases consisted of four in-hospital diagnoses and ten out-of-hospital diagnoses. For clear explanation, we revised the figures and manuscript.

2. This paper conclude hospital shutdown for extended period was not necessary and testing of all hospital staff and inpatients without risk assessment are time-consuming. However, the paper does not provide enough scientific evidence how this conclusion is reached. Low confirmed probability ex-post is not the evidence you can use since the risk of outbreak is still exist. Just explaining the action taken by Korea is meaningful.

: We fully agree with the comment and revised the manuscript. The revised conclusion is as follows. “In conclusion, the robust control of the COVID-19 outbreak further minimized the transmission of SARS-CoV-2 in the hospital and local communities. However, there was also a debate over the appropriate period of hospital shutdown and testing of all hospital staff and patients. Future studies are required to refine and establish the in-hospital quarantine and de-isolation guidelines based on the epidemiological and clinical settings.”

---

## [Editor Report · Decision Letter 1]

3 Aug 2020

Containment of a healthcare-associated COVID-19 outbreak in a university hospital in Seoul, Korea: a single-center experience

PONE-D-20-13049R1

Dear Dr. Lee,

We’re pleased to inform you that your manuscript has been judged scientifically suitable for publication and will be formally accepted for publication once it meets all outstanding technical requirements.

Kind regards,

Xia Jin, MD, PhD

Academic Editor

PLOS ONE
---

## [Editor Report · Acceptance letter]

7 Aug 2020

PONE-D-20-13049R1 

Containment of a healthcare-associated COVID-19 outbreak in a university hospital in Seoul, Korea: a single-center experience 

Dear Dr. Lee:

I'm pleased to inform you that your manuscript has been deemed suitable for publication in PLOS ONE. Congratulations! Your manuscript is now with our production department. 

Kind regards, 

on behalf of

Dr. Xia Jin 

Academic Editor

PLOS ONE